# Living Well Together in a Climate-Changed Future: Religious Imaginaries on the Cutting Edge of Genetic Technology

## Lisa H. Sideris

Environmental Studies Program/Affiliate Department of Religious Studies, University of California, Santa Barbara, CA 93106, USA; lsideris@ucsb.edu

**Abstract:** This essay focuses on the emotional and relational investments of scientists and others engaged in and supportive of genetic technologies used in conservation efforts, with particular attention to the different moral and religious imaginaries that fuel endeavors to save species threatened by climate change and extinction. I argue that two distinct visions and competing religious repertoires can be discerned in the secular landscape of genetic technologies deployed in coral restoration and de-extinction. Each endeavor brings forth its own forms of magic, myth- and meaning-making. At the heart of coral protection is the symbol of the holobiont, suggestive of cooperative endeavors, collective labor, networking, and distributed and embodied knowledge. Central to de-extinction imaginaries are motifs of individual competition, machine metaphors, "selfish" genetic components, and a spirit of entrepreneurial excitement and profiteering. The essay contrasts these two visions as competing accounts of relationality—or the lack thereof—and asks which religious and moral imaginaries we should embrace as we move into an era marked by intensified technological intervention and high-risk efforts to address the effects of climate change. I suggest that the values that drive de-extinction technologies are largely at odds with environmental and social goals of living well together, as humans and more-than-humans, in a present and future world transformed by climate change and species death.

**Keywords:** corals; de-extinction; extinction; holobiont; religious imaginaries; genetic engineering; gene drives; climate change; Ruth Gates; George Church; Stewart Brand; Kevin Esvelt





## 1. Climate Emergency and the Genomic Turn

As I write, in late summer 2023, the drumbeat of climate change is growing too loud and insistent to ignore, for all but the most committed denialists. An exceptional marine heatwave has warmed the North Sea to temperatures not seen in over 170 years. In Southern California, where I live, scientists are tracking a rare and powerful hurricane that is spinning toward the coast, fueled by anomalously warm Pacific waters. A storm of this sort has not made landfall in California in nearly 84 years. In Florida, a massive coral bleaching event caused by record-high ocean temperatures is currently underway. Meanwhile, on land, the town of Lahaina, Hawai'i, with thousands of years of rich history, has just been devasted by wildfire, leaving a path of destruction that eclipses all previous records. In Canada, wildfires have recently burned an area the size of New York State— easily the worst wildfire season on record for that part of the world. Smoke from the fires has choked cities across the U.S. and beyond, all summer long. A blanket of smoke is hovering over the Pacific Northwest as well, from wildfires on both sides of the border. The dismal list goes on and on. These days, the weather *is* the news, and "unprecedented" is beginning to lose its meaning.

While headlines often focus on human losses, the impacts of these events on wildlife and ecosystems are also grave and rapidly intensifying. Earth has witnessed numerous mass extinctions over the course of its long history, but the current extinction event, often called the sixth mass extinction, is unique in that it is driven by human activities including

unsustainable use of land, water, and energy, and climate change. For some species it is already too late, or nearly so. Indeed, for many ecosystems around the world, the idea that we can return nature to a pristine state looks increasingly fantastical. Consequently, some scientists are turning to cutting-edge genetic technologies to rescue the remaining members of some species from extinction or to attempt to reverse extinction once it has occurred. The latter approach is referred to, somewhat misleadingly, as de-extinction. Both projects—rescuing coral and de-extincting—are swept up in the "genomic turn" that is reshaping much of contemporary environmental management (Braverman 2018, p. 201).

The emotional and psychological impact of these mounting crises on climate scientists and conservationists is receiving wider attention, as more and more researchers speak candidly about the difficulties of continuing their work in the face of grim odds and rapidly vanishing creatures (Einhorn 2023). The affective register of research on climate change and extinction often varies widely across different types of conservation and restoration projects, ranging from feelings of profound grief and despair to excitement and optimism about the life-saving and life-shaping power of emerging technologies. In what follows, I explore these dynamics along the lines of what ethnographer and legal scholar Irus Braverman calls the "emotional and relational landscape" of researchers who are using genetic tools to address the impacts of climate change (Braverman 2017, p. 56). These varying landscapes hint at how individuals understand themselves to relate—or not—to other lifeforms and larger ecological and social contexts. They offer important insights into the justifications, values, and motives that drive the deployment of technologically intensive approaches to saving species threatened by climate change. In short, they speak to what religion scholar Lori G. Beaman calls "the ability to imagine oneself in relation to others and what one has in common with others" (Beaman 2017, pp. 11–12).

Relational sensibilities, or the lack thereof, are often expressed through affective discourse that attends contemporary environmental management practices. But affect is not the whole story, for what is revealed in these imaginaries is a construal, or perhaps a reconfiguration, of one's place in the world that is shaped by and expressed through science and technology. Put differently, scientific and technological understandings and objectives are inseparable from affective commitments to intervening in nature (Schaefer 2022). These interventions may take the form of world repairing—work that brings diverse constituencies together to make the planet "liveable for human and other-than-human beings"—or world-dominating, or perhaps some complex combination of the two (Beaman and Stacey 2021, p. 1).

My claim is that these "construals" of oneself in relation to others and the wider world (both social and ecological) constitute religious imaginaries.[1] "The power of the term imaginary", Beaman and Stacey argue, lies in its "ability to traverse distinctions between religious and nonreligious ways of understanding the world while avoiding thinking of either as unified systems" (Beaman and Stacey 2021, p. 4). Imaginaries are often conveyed through, or partially comprised of, what Timothy Stacey calls religious repertoires: images, symbols, stories, and rituals. They stake a claim to who or what "the human" is or ought to be; they are bound up with meaning-making, and sometimes myth-making. Religious and moral imaginaries form a loose "constellation" of stories, events, and characters (real or fictitious) that influence how one acts in the world, and how one envisions possibilities and renders action meaningful (Stacey 2022, p. 81). Practitioners of emerging technologies used in restoration or de-extinction express their visions of life in the course of applying sophisticated tools. While the tools themselves may be similar or even identical, the worlds these practitioners hope to bring into existence contrast in important ways with one another in the cases we will examine below.

In settings and activities that appear devoid of religion, as with certain kinds of sustainability work, we can nevertheless identify religious repertoires—"practices that make some ways of perceiving the world meaningful and others meaningless: what human and other beings are; what the world is; how these interrelate; and whether things could or should be different" (Stacey 2023). Often, these visions partake of forms of "magic" or

magical thinking with objects or technologies that make another world seem possible, even (perhaps especially) when the odds are strongly against it. Magic here does not "compete with science but complements it" (Stacey 2023, p. 117).

As Stacey's studies of activism show, magical thoughts and feelings often shape sustainability work, including environmental activism which "involves taking on causes that all the evidence suggests are impossible to win or else already lost" (Stacey 2022, p. 124). For those working to save corals, for example—an effort that merges science with activism to a degree not often seen in scientific endeavor—magical imaginings occur with some frequency, as do enchanting underwater encounters with corals themselves. Similarly, scientists working to bring species back from extinction also engage in forms of magical thinking and "against-all-odds" efforts to call forth new worlds through science and technology. Techniques of de-extinction and related genetic technologies are also spoken of in terms that express awe at genetic elements and the machine-like bits and pieces of organisms with which these scientists tinker as they experiment with the basic ingredients of life.

These religious repertoires warrant close examination because of the power they hold for effectively "locking in some socio-ecological futures and locking out others" (Stacey 2023). What forms of magic—what kinds of religious repertoires and imaginaries—do we want to preserve and pass on as humans collectively enter a time of extreme climate disruption and intensified intervention in nature? Which religious visions are most conducive to living well together in a climate-disrupted present and future? (Beaman 2017). In the regulatory void that presently exists for many emerging genetic (and other) technologies, scientists are increasingly left to self-regulate—or not. In this setting, the "values and visions"—the moral imaginaries—of individual scientists take on enormous importance since they will "at least partly determine the scope of the research" and its "normative dimensions" (Braverman 2017, p. 56). At the heart of this essay is an examination of competing religious visions and religious repertoires of scientists experimenting with genetic technologies to safeguard and recover species from conditions of extreme climate change and extinction. Two widespread and seemingly similar forms of high-tech socio-ecological intervention are closely scrutinized and disentangled, drawing on tools and critique from the study of religion.

## 2. Emerging Environmental Technologies: Similarities and Differences

Two broad examples of technologically-intensive forms of management—scientists working to save coral reef ecosystems from climate-induced extinction, and those adopting genetic technologies to undo extinction—seem to mark a departure from traditional environmental approaches.[2] They offer a rich comparison owing to numerous apparent similarities as well as what I see as their significant ethical differences. The juxtaposition also raises questions about the usefulness of categories like synthetic biology or genetic engineering that are often invoked to classify (and possibly denounce) certain types of interventions or to distinguish them from others. Let us look first at the similarities.

De-extinction efforts, like the proposed resurrection of the wooly mammoth, are sometimes framed as a form of climate mitigation (for reasons explained below), which suggests a fruitful comparison with genetic technologies deployed to adapt corals to conditions of climate change. Additionally, both coral-saving and de-extinction technologies are high-profile initiatives that attract rapt and sometimes sensational media attention, as well as the patronage of billionaire investors. Researchers involved in both endeavors often attain a kind of celebrity status as they come to embody the hopes and fears of people all around the globe in a time of great climate peril. Celebrity status creates a magic of its own, as these scientists are mythologized in award-winning documentaries and feature films.[3] Both projects constitute a kind of last-ditch, by-any-means-necessary form of conservation and both carry with them an uneasy (to some) sense that we are transitioning from the familiar territory of environmental protection into a brave new world of biopolitical manipulation. As such, these interventions are part and parcel of a new landscape of Anthropocene ethics,

where human oversight and intensive technological involvement in nature define a new normal (Kolbert 2021).

Both forms of environmental management draw upon technologies that might be categorized as synthetic biology. There is no single, agreed-upon definition; rather, synthetic biology consists of a suite of applications from which researchers select. The National Human Genome Research Institute describes synthetic biology as a "field of science that involves redesigning organisms for useful purposes by engineering them to have new abilities" (NIH n.d.). Whether or not a particular technology is seen to align with the description of synthetic biology may have more to do with how words like "natural", "novel", "artificial", "engineering", and "useful" are defined than with the technologies themselves. Terms like useful are also vague. Is the usefulness of the technology understood in anthropocentric terms or is it instrumental in the survival of organisms themselves? Does it matter?

Another account of synthetic biology, offered by Harvard geneticist George Church whose work is at the center of contemporary discussions of genetic technologies like CRISPR, defines it as "the science of selectively altering the genes of organisms to make them do things that they wouldn't do in their original, natural, untouched state" (Church and Regis 2012). Yet, like the previous definition, this raises many additional questions. What constitutes a natural, untouched state at a time when the entire planet is impacted by climate change? What "things" are organisms being made to do? Moreover, management approaches that involve these technologies are typically pursued precisely because the "natural" state of an ecosystem has so radically shifted that further intervention seems warranted. In this context, Elizabeth Kolbert's recent work explores the "recursive logic" of the Anthropocene, whereby human interventions and alterations of the environment necessitate *further* rounds of interventions to address the negative impacts of the initial, flawed intervention—potentially, in an iterative fashion (Kolbert 2021, p. 117). Thus, for example, one response to widespread coral bleaching and death caused by warming and acidic oceans is to manipulate the genes of corals to make them more resilient to climate change. Humans can assist their evolution, essentially augmenting and speeding up adaptation to climate change.

There are currently a handful of techniques for assisting coral evolution. These include stress exposure or "stress tests" that allow some corals to become acclimated to warming temperatures (changes may then be passed on to the next generation); selective breeding that puts the sperm and eggs of the most resilient corals together to produce super-performing corals; methods of assisted gene flow, also called assisted migration, that spread beneficial mutations around coral populations by physically moving them (e.g., translocating coral stock with heat tolerance from warmer to cooler locations that will be heating up); and techniques focused on the photosynthesizing plant cells within corals—or the microbiome, as it is called—to optimize the symbiotic relationships that keep corals healthy.

We might ask then: are techniques of assisted evolution that expedite "natural" resistance to heat in corals (or their algal symbionts) a form of synthetic biology? Do they constitute "engineering"? Does selecting for heat resistance, in and of itself, amount to synthesizing something "novel" or is it merely enhancing a pre-existing natural ability? Are assisted corals doing "things" they (sometimes) do in "nature" or not? The answers are not crystal clear (and indeed, some coral scientists vehemently deny that they are engaged in the engineering or synthesis of anything "new"). Some might argue that there is an obvious difference between these coral restoration techniques and de-extinction in that the former seek to "save" while the latter aspires to "bring back" organisms that have met their demise. Moreover, one might say, it is no wonder that these two efforts lend themselves to different emotional registers and moral imaginaries, given that scientists in the first camp form affective bonds with the creatures whose deaths they are desperately trying to prevent, whereas those in the latter group often have no direct experience with creatures they hope to "resurrect".[4]

However, the difference between "saving" and "bringing back" is murkier than it might seem. To see this, consider again the example of enhancing coral resilience through assisted evolution. One way to describe assisted evolution is to say that it essentially *remakes* corals in ways that better align them to a human-imposed target, namely, current and future anthropogenic conditions of climate change. "In moving species or ecosystems toward states that they have never been in before, we are in some senses newly defining what we deem to be valuable and using the tools of assisted evolution to *create* value" (Filbee-Dexter and Smajdor 2019, p. 3).

Similarly, de-extinction projects entail the creation of novel organisms from existing genetic materials that are combined with fragmentary (or "ancient") DNA culled from specimens of extinct organisms, producing what is at best a proxy of the creature that went extinct.[5] In some cases, the genome of this new creation is deliberately tweaked to make it a better fit with environmental conditions humans have created. For example, owing to a dearth of intact DNA for most extinct organisms, efforts to de-extinct creatures like the wooly mammoth combine genetic material from closely related Asian elephants with DNA extracted from specimens. The resulting organism would be a hybrid creature, a mammophant, that looks like a mammoth and has some mammoth-like properties like cold tolerance. Mammoths have been extinct for at least 4000 years (and much longer for some populations). In order to give an environmental patina to the mammoth de-extinction project, researchers often cast it as a conservation effort. By editing wooly mammoth genes into existing (endangered) Asian elephants, they can create a creature that is more cold-tolerant than its natural counterparts, and thus able to live in habitats not normally suited to them (Wray 2017). Expanding elephants' range to colder regions is one way of addressing a major (human-caused) threat to their survival, namely, habitat fragmentation and encroachment.

In short, because de-extinction is not the "return" of what was lost but rather the *created* approximation of that thing (a creature subject to further refinements as deemed necessary), this technology is not so dissimilar from some forms of coral "restoration" as it might appear. Arguably, both define a new target set by humans; both, one might say, *create* value. While the elephant case differs from that of corals in that genes are imported from a different species, both produce organisms better adjusted than their erstwhile "natural" counterparts to human-caused conditions, including projected *future* conditions.

In conclusion, then, given the lack of clarity or consensus regarding terms like synthetic biology and genetic engineering, and the surface similarities between coral restoration and de-extinction, perhaps in differentiating them, we do well to focus less on the tools involved than on the intentions, motivations, and imaginaries of the people using them. What do they understand themselves to be doing? What ultimate ends are being pursued, and why? These ultimate goals, these modes of self-understanding, are inextricably linked to ethical and affective states, and to broader values and worldviews. Put differently, the label given to a particular technology seems less informative, and less revealing, than the moral, emotional, and religious imaginaries that embed these tools and drive their deployment.

As we, as a society, grapple with questions of whether and how to use these technologies, it is critical that we scrutinize these broader visions of relationality and consider the kinds of worlds they seem likely to usher into existence, both for our own sake and the sake of myriad other living creatures. In evaluating these projects, we need to consider factors beyond what the technologies do at the level of genetic manipulation. One place to start is with the religious imaginaries that orient and give meaning to these high-tech interventions.

## 3. An Orientation to Two Competing Imaginaries

For all of their similarities, coral rescue and de-extinction invite reflection on two competing accounts of what it means to be human, about how we enact or deny relationships with other organisms through innovative technologies, and even how to define life itself. The broad contours of two visions might be articulated as follows: one approach, evident among prominent coral scientists, tends toward a communitarian ethos of human

and nonhuman life. It understands technology as guided and potentially constrained by relationships and values inherent in nature itself and, therefore, worth safeguarding, even as tools of environmental management increasingly coopt and refine natural processes. A second approach, recognizable in some prominent de-extinction efforts, enshrines in various ways principles of individualism and individual (human) creativity, and social and evolutionary competition. It suggests humans as architects of life, creatures who confer intelligence and meaning to natural processes.

I am not suggesting that we view these two visions in stark, categorical terms, such that the coral community adheres, monolithically, to one ideology while proponents of de-extinction are wholly committed to the other.[6] Both, in fact, share a general optimism regarding the potential of technology in the realm of conservation. However, the basic elements I have sketched out in each case are clearly discernible as comprising two types, and their comparison prompts important questions regarding the attitudes that attend human management and manipulation of the environment.[7]

We begin with an overview of coral imaginaries. Corals and coral reefs are remarkably rich in meaning and symbolism, and they are central to a movement to think differently about the very nature of life and how living beings relate, and should relate, to one another. In an age of extinction and environmental precarity, corals appear to hold great significance as symbols of relationality and collectivity—representing some of the values humans might cultivate in order to address societal factors driving climate catastrophe.

## 4. Coralation: Corals in the Human Imagination

"There's a whole mutualistic vein to this that affected my psyche. I believe that there are great lessons there for every process we engage with . . . I mean, mutualism is where we should be going—we should be trying to balance our relationships on this planet".

–Ove Hoegh-Guldberg, professor of Marine Science, in conversation with Irus Braverman

For much of human history, corals have played an important role in the imaginaries of many cultures across many geographic regions. Today, at a time of great environmental peril, they are "revelatory figures with whom we may think through, and feel, our contemporary social and ecological vulnerabilities" (Braverman 2018, p. 249). It is a sad irony that just when scientists and the broader public are beginning to grasp how singular these creatures are, many coral reefs are in rapid decline. Widespread coral "bleaching events" caused by warming oceans were once exceedingly rare, but are becoming more and more frequent and severe.[8] It is estimated that by 2030, more than 90 percent of the world's coral reefs will be threatened by human activities, including human-caused climate change and ocean acidification. By 2050, scientists estimate that nearly all reefs will be threatened, with 75% facing high, very high, or critical threat levels (Coral Reef Risk Outlook n.d.). Those seeking to arrest their downward spiral often engage in forms of myth-making and magic that have long defined the human-coral relationship.

Corals share a symbiotic relationship with photosynthesizing algae, called dinoflagellates, that live within them and serve as a source of food. Bleaching results, it is thought, when the symbiotic algae in coral polyps are unable to photosynthesize properly and are consequently ejected by the host. Most corals cannot survive without the products of photosynthesis the algae provide. Polyps become translucent, appearing as white bone.

The symbiotic algae-animal relationship is a steady source of fascination with corals, for it calls into question the stability of the plant/animal boundary (and even the animal/plant/mineral distinction). A key principle of coral life is what Braverman cleverly terms coralation, a neologism that expresses material interconnection and symbiotic interrelation, while also signifying a shapeshifting quality that makes corals difficult to pin down. Strange, wonder-inducing creatures, corals have long confounded efforts to classify and taxonomize them. To some, they suggest the irreducibility of life to the status of a mere

specimen (Navakas 2023a). An assemblage of stone and flesh, they also invite speculation on the line between life and nonlife (Helmreich 2016, p. 49).

We humans have long been captivated by corals, despite their lack of humanlike and mammalian qualities that typically attract us—no endearing face or eyes, no limbs to speak of, no clear sex. Not even a brain. To be sure, corals can be beautiful with vibrant rainbow or blood-red hues, but compared to many charismatic "poster species" of the environmental movement, they are alien and difficult to anthropomorphize. Adding to their classificatory complexity, corals are not individual creatures, nor are they even a pair of organisms, but something like a community or complex consortium of organisms living together.[9] As assemblages, superbeings, or "holobionts"—a term recently popularized by coral scientists and a wide array of scholars—corals inspire an appreciation of how even seemingly discrete entities are in fact deeply entangled with and constitutive of other living beings. Multiple layers of meaning, many of them hopeful, are "encapsulated within the corporeality of coral life" (Braverman 2018, p. 21).

The question of what, if anything, constitutes a coral "individual" is so confounding that it creates difficulties in applying environmental law to them. Ascertaining their status as endangered or threatened requires that individual corals be "both identifiable and quantifiable", a task ill-suited to these creatures (Braverman 2018, p. 206). While corals may be vexing from a legal standpoint, they have fired the imagination of many scholars in the environmental humanities for whom they representant distributive modes of agency and subjectivity (Helmreich 2016, p. 54). For all of these reasons and many others, corals are and always have been, "good to think with" (Braverman 2018, p. 11).

In popular culture, religious mythology, and as a matter of scientific fact, corals are life-giving, world-making creatures. They are widely prized for ornamentation and medicine and for the wisdom they impart for living cooperatively with others. Reef-building corals create elaborate, living infrastructures that benefit a wide array of creatures and support an astounding degree of diversity, providing homes and sustenance for at least 2 million other species, or approximately a quarter of all life in the oceans. Humans too depend on corals which protect coastal areas from storms and erosion, and provide sources of food and medicine, as well as many recreational activities and tourism dollars.

Modern-day scientists and conservationists who are working to save them are participants in the creation of coral imaginaries, no less than ancient cultures who spun stories of the life-sustaining and healing power of coral. Corals gathered from the sea have long played a role in religious rituals, offerings, and ceremonial architecture in Pacific regions (Molle et al. 2023). Among Indigenous Pacific Islanders corals are associated with genealogy and the origins of life itself. According to the Kumulipo, the Hawai'ian creation chant, the coral polyp was the first organism created, along with the first man and woman. The message of the creation story is that "life in the sea and life on land are inexorably connected, and what we do on land has a direct connection and impact on all organisms in the sea" (Coral Reef Alliance 2016). In other parts of the world, corals have been invested with protective and talismanic powers. "From the Middle Ages until the 19th century, anxious new parents across Europe and North America clasped red coral necklaces and bracelets to their children's bodies . . . because coral symbolized physical and spiritual protection" (Navakas 2023b). In paintings from the 19th century, corals—especially red corals—often represent fertility, family, blood, and bloodlines. They are also broadly associated with labor and laboring bodies of various sorts, including women's labor in childbirth (Navakas 2023a), a point we will consider again.

The Great Barrier Reef which spans 1600 miles off the north-east coast of Australia is large enough to be seen from space—an astonishing feat for a brainless plant-animal amalgam. In view of their world-building powers (and despite their non-anthropomorphic qualities), corals are also seen to exhibit a variety of capacities often associated with humans. They are routinely hailed as architects, designers, manufacturers, and even chemists capable of producing anticancer or antiviral drugs, as well as their own sunscreen to shield themselves—within limits—against conditions of climate change (Berwald 2022). While

arguments for coral protection often cite a litany of anthropocentric interests and ecosystem services, many researchers speak of their value and uniqueness in terms not easily reduced to a utilitarian framing. Words like "generous", "hospitable" and "beneficent" are often applied to them.[10] Various coral-induced epiphanies have reconfigured the landscape of the biological sciences, lending support to a "rhizomatic" perspective that favors nonhierarchical and nonbinary categories and modes of thinking.[11] In short, corals and coral-like assemblages have the potential to challenge theories of the autonomous individual both in nature and in society. Ove Hoegh-Guldberg, the coral scientist whose appreciative remarks about mutualism are quoted above, observes that a significant factor in the destruction of the Earth is the still-prevalent Victorian notion of "survival of the fittest" which exalts the individual and individual success (Braverman 2018, p. 60). Corals test the limits of individualist imaginaries and assumptions.

Disdain for competitive individualism and praise for symbiosis are not uncommon sentiments among those who study and protect corals. In the world of biology, the symbiotic view of life is often opposed to gene-centered neo-Darwinian frameworks that reify the individual and promote competition as the driving force in a zero-sum game of evolution. In this sense, corals act as "a constant reminder of the importance and pervasiveness of collaboration and mutualism", as Braverman writes, and "they have thus been central to the recent scientific realization that 'we have never been individuals' and that 'we are all holobionts'—assemblages of microbial forms of life with complex interrelations" (Braverman 2018, p. 249). In their now-classic work on this subject, *What is Life?*, biologist Lynn Margulis and eco-philosopher Dorion Sagan understood symbiosis to undermine "the prevalent notion of individuality as something fixed, something secure and sacred". Extrapolating from this account, the human has also come to appear more like a composite than a single entity, Margulis and Sagan argue, "as each of us provides a fine environment for bacteria, fungi, roundworms, mites, and others that live in and on us" (Margulis and Sagan 1995, p. 236). Coral lifeways have helped to inaugurate a shift toward conceiving of value, even sacredness, as inhering in collectivities and in relationality itself.

## 5. Coral Magic

For many who study corals, the magical quality of corals, and the enchantment of undersea life generally, orients and sustains their work. The experience of entering alien ocean worlds—both literally and imaginatively—forms part of the religious repertoire and moral outlook of many coral scientists. Proximity to this alien world, and empathic sensibilities cultivated through repeat, ritual-like encounters with otherness, may have something to do with the degree of care and commitment demonstrated by many coral scientists.[12] The meditative experience of diving is part of what draws some to a career in marine science and keeps them coming back to a majestic world normally hidden from human sight, even as repeat visits bring despair over the deteriorating condition of corals. For some, diving is a spiritual practice of being present to one's surroundings and oneself. "The background noise of terrestrial life is muted and one automatically attunes to the breath . . . Inhaling takes you up, exhaling brings you down . . . Time is defined by the movement of the breath" (Braverman 2018, p. 153). Even among scientists whose work is conducted largely in laboratory settings, encounters with life in the ocean world are what initially "hooked" them and drew them to their work. Ruth Gates described "falling in love" with corals during an early diving expedition in the West Indies (Fuji 2016). Though her work involved extensive lab time, Gates was forever finding excuses to head back into the water or to stay there longer than was strictly necessary. In an interactive documentary about corals called "Lost Cities", released just after her untimely death in 2018, Gates evokes the magic of the dive. You roll off the side of the boat into the water; blue bubbles erupt all around you. As they disperse, a strange vista unfolds beneath you, an "underwater cathedral" of corals. "How", Gates asks in the voiceover, "can nature even do something like this?"

Similar expressions of love and affection for corals, and wonder at their mysterious qualities, are common among researchers, as are experiences of intimate communion with coral life. While some scientists have to be coaxed to speak openly about personal and spiritual investments (having been disciplined throughout their career to bracket personal feelings), many express intense emotional involvement with the creatures they study. Note, for example, one exchange involving marine biologist Les Kaufman, in which he references a special form of communion between corals and those working to keep them alive. "We're listening to the corals", Kaufman tells Braverman, "this is how they talk." "So the corals are whispering?" she asks. Kaufman explains, "Coral whisperer means I'm whispering to the corals. But the coral is whispering back" (Braverman 2018, p. 1).

"Coral Whisperer" or "Reef Whisperer"—terms denoting individuals with unique abilities to communicate with nonhuman lifeforms—are monikers often attributed to people trying to save them. A graduate student coral whisperer at Boston University describes her research on assessing corals' ability to withstand stress. The work entails wounding a coral polyp by scraping off a tiny bit of tissue and then monitoring how the corals heal under varying environmental conditions. "I feel a little bad about it", the student, who describes herself as vegetarian, confesses to a reporter. She speaks "like a loving pet owner" as she feeds her rock-like charges with a slurry of shrimp administered with a turkey baster. "It's pretty cute", she admits. The lab space where the experiments take place is affectionately dubbed "The Room of Requirement", with a nod to the special chamber that magically appears in times of urgent need, in the Harry Potter series (Barlow 2014). In her conversations with Braverman, Gates again turned to a familiar sacred image: "Coral reefs are my cathedral . . . I have a deep sense that this is where I am meant to be". Expressions of love and fascination, she acknowledges, are "all the wrong things for a scientist to say" (Braverman 2018, p. 232).

Intimations of magic and myth abound. One marine biologist describes his connection to corals as follows. "I'll start with magic", he says, referencing the ancient Greek myth in which Perseus kills Medusa, spilling blood that gives rise to red corals. The biologist eagerly awaits the full moon, the time when the corals spectacularly spawn. Upon discovering newly spawned larvae in the aquarium where he had placed mother colonies, he says, "I feel like they are my babies" (Braverman 2018, p. 17). Another researcher kisses the aquarium that holds her "beloved" specimens. "I created such a strong bond with them", she explains. Working with them was a "dream that came true". Describing how she would race to the lab at all hours of the night when a crisis emerged or equipment malfunctioned, she says "I loved it!" When they spawned, she adds, "I was the happiest person in the world" (Braverman 2018, p. 254). Like Rachel Carson who insisted on returning sea creatures to their homes after observing and sketching them, dedicated researchers carry coral fragments used in the lab back to the ocean, in the hopes that they might reattach themselves.[13]

More generally, for many who study and protect them, corals provide a model of, and for, the interconnection of life and the importance of collaborative endeavor. As noted previously, they are also broadly associated with labor and laboring bodies of various sorts, including women's labor in childbirth (Navakas 2023a). Massive coral reefs inspire the hopeful idea that many laborers working together can create something grand and long-lasting, as reefs are thought to expand by sustaining others rather than displacing them (Navakas 2023b). They are a prototype of smart growth.[14]

These forms of coral meaning-making are frequently voiced even, or especially, among those pursuing assisted evolution technologies. This may seem surprising, given that Gates' dedication to creating "super corals" through selective breeding and expedited evolution seems the very emblem of a survival-of-the-fittest approach. (The "super coral" designation seems to have originated with media reports). "Just as elite athletes are selected and groomed from a young age to rise above their competitor", as one journalist breathlessly describes the project, "in this lab hundreds of juvenile corals are being conditioned for a showdown of survival of the fittest" (Mascarelli 2021). Gates herself invoked the athlete

metaphor to describe super-performing corals. Yet, at the same time, it is mind-boggling to extend the individualist framework of "the fittest" to creatures who not only defy classification as individuals but are variously categorized as metaorganisms, hybrids, and even chimera (consisting of two or more individuals). Gates often stressed that other creatures are heavily involved in the making of a high-performing coral; they do nothing alone.

Gates' own interpersonal style and sensibilities resonated with the cooperative style and distributive agency of the creatures she studied. She was a vocal critic of the siloed, ego-driven, individualist ethos of academia, describing the system as essentially broken. Creative problem-solving, she believed, was best nurtured in collaboration with others. Science writer Ed Yong, in an eloquent remembrance of Gates, makes a similar connection: "Reefs enrich the oceans by creating spaces in which thousands of diverse species can thrive. Gates nurtured a vast community of researchers by opening doors for them and supporting their lives" (Yong 2018).

Others too have noted how the symbiotic, assemblage-like nature of corals is mirrored in networked approaches to saving them—efforts that often blur the boundaries between scientific fact and cultural values. As Braverman shows, the relative optimism of scientists doing restoration work with corals springs from the sense of hope created in "human-nonhuman networks and collaborations" (Braverman 2018, p. 5). Gates went against the grain of conventional scientific practice in turning her research into hopeful action. Writer and social designer Cheryl Heller argues that Gates routinely harnessed "the power of relationships and a shared vision" to accomplish her goals, implementing principles of "collaboration and net-worked cocreation". She constructed a social architecture that mimicked the coral lifeforms she regarded as the "genius architects of the natural world" (Fuji 2016). Those networks included not just other scientists and students, but an assortment of devoted conservationists, politicians and government agents, Indigenous people, filmmakers and journalists, and even schoolchildren (Heller 2018). The "spirit" of collaborative undertakings across diverse constituencies is a significant source of the magic that draws people to activist endeavors, as Stacey (2022) argues. Gates excelled at evoking spirit among coral scientists and caretakers, in the face of almost impossible odds.

## 6. Hopeful Labor

Admiration and advocacy for corals extend well beyond scientific circles. The collaborative spirit of coral protection and reverence for the collaborative essence of coral lifeways are particularly strong among networks of women researchers and artists. Gates described her mission-centered approach as "feminine", noting the marginalization or dismissive attitude women in the field experience in the male-dominated science community, where women's approaches are seen as "emotional". "I constantly talk about my passion for coral reefs, my emotional connection" (Braverman 2018, p. 232). While women remain underrepresented in marine sciences, as they do in STEM fields generally, many notable leaders in coral reef conservation, including some conducting trials with assisted evolution, are women. In fact, women have made landmark contributions to coral conservation for decades (Foxwell-Norton et al. 2021) and many of the traditional and current custodians of coral reefs are Indigenous women.

A case in point of the gendered dimensions of coral protection is the Crochet Coral Reef project. Feminist scholar of science and technology studies Donna Haraway celebrates corals as symbiotic (or sympoetic) creatures, arguing that they represent a counterpoint to entrenched patterns of Western thought that exalt "human exceptionalism and the utilitarian individualism of classical political economics" (Haraway 2016, p. 57). Haraway's interest in coral lifeways led to her involvement in the Crochet Coral Reef, described as an "ever-evolving nature-culture hybrid" that blends art, applied mathematics, feminism, evolutionary theory, environmentalism, and community-based practice. Created by Australian sisters Margaret and Christine Wertheim, the project is a "one-stitch-at-a-time meditation on the Anthropocene" (Crochet Coral Reef n.d.).

The crochet reef exemplifies the longstanding association of corals with collective labor, a subject explored in fascinating detail by Michele Currie Navakas in *Coral Lives: Literature, Labor, and the Making of America*. Captivating in its own right, the crochet project calls into being a model of what humans can achieve when they work together, like so many coral polyps, in ways that neither ignore the severity of ecological problems nor give in to "fantasies that rescue is around the corner from some sudden technological solution" (Crochet Coral Reef n.d.). It is not clear what sort of technological solution would be considered "sudden" or fantastical by the reef's main creators, but the hope expressed by those involved in the crochet reef mirrors the discourse of women coral scientists who engage in assisted evolution technologies as an expression of hope. Those who turn to these technologies might be seen as less "hopeful" to some, in the sense that they believe corals can no longer recover on their own; theirs is a kind of climate realism mixed with hope. In rejecting fantasies of technological rescue, practitioners of coral crochet narrate hope in terms that resonate with the orientations of coral scientists, even as the latter reach for technological solutions. In particular, hope often lies in, and is activated by, the collaborative networks of practitioners.

The crocheted creations intentionally mimic the structure of reef-like forms, drawing on techniques of hyperbolic (non-Euclidian) crochet, a type of algorithmic weaving that utilizes a surface that exhibits negative curvature. In an essay that explains the hyperbolic geometry behind the crochet project, Margaret Wertheim remarks on the power of corals and sea slugs to test the limits of human abilities and imaginations through their bodies and creations. Corals "who'd never studied non-Euclidian geometry had meanwhile just been doing it" (Wertheim 2016). Elsewhere Wertheim articulates what is at stake in the project, namely, the value of *embodied* knowledge (Wertheim 2009). Wertheim notes that we live in a society that valorizes symbolic forms of representation—algebraic representation, logic, equations, codes—over embodied modalities. Yet, nature has been generating hyperbolic shapes for hundreds of millions of years, while mathematicians denied that such a thing was impossible, in part because they failed to pay attention to the living world around them. "Does a sea slug 'know' hyperbolic geometry?" The Wertheim sisters ask. "Does a head of coral?" The project ventures that "in some sense they do". In making these structures, corals are *doing* math, the sisters argue (Crochet Coral Reef).

The crochet project advances its agenda along two related fronts: calling attention to forms of intelligence embodied in corals that are now threatened with climate change, and celebrating the sophistication of "domestic feminine art", a parallel form of embodied knowledge—and play—enacted by thousands of (mostly) female weavers. The discourse surrounding the project elevates feminist practice, while also expressing deference to, even reverence for, nature's own ways of knowing and doing. It provides an opportunity to "affectively attune ourselves" to the grave conditions of corals, without giving in either to despair or naïve optimism (Davis 2020).[15]

The "evolutionary" vision embedded in the crochet project is a nod to the many reefers (up to 20,000 participants) who make their own satellite reefs. As the project site explains, the collaborative spirit challenges prevailing notions of creativity as the purview of the *individual*, much as corals themselves have enabled a paradigm shift away from a neo-Darwinian focus on the individual (per Margulis and Sagan's critique) as something fixed and sacred. In other words, the project evokes a distinctly cooperative style of evolution that one might not immediately associate with Darwinism in its competitive, "red in tooth and claw" mode. "By inviting in people from all walks of life the *Crochet Coral Reef* offers a radical alternative to the model of artist as singular prodigy" (Margaretwertheim.com n.d.). One might say, it provides a vision of flourishing together.

Crucially, as a scholar close to the project argues, the reef also offers an alternative vision to that of technological rescue executed by the lone heroic scientist. It enshrines:

> the power of collective action, as an embodied resistance to the damaging narratives of the male genius who will come up with the techno-fix to solve the current ecological crisis. Instead it represents a global insistence of the possibilities of

collective dispersed action without the need for a hero. It is this politics, learned from the watery depths and dedicated to feminist praxis, that is so needed at this moment in time (Davis 2020, p. 73)

A similar spirit animates the efforts of many coral scientists who cultivate realistic hope and reverence for nature, even as they engage with ever-more interventionist approaches to saving corals. Some of the most optimistic of coral scientists—Gates among them—speak in measured tones about the prospects of technological rescue and the enormity of the task of saving reefs from the ravages of climate change. Gates often called attention to the "mystery" of corals—their baffling ability to live essentially forever, the inability of science to fully grasp all that corals know and can do. In a similar vein, an Australian scientist working to save the Great Barrier Reef through "industrial-scale" interventions that range from marine cloud brightening (a kind of small-scale geoengineering) to assisted evolution and genetic engineering, remarks, "It's just absolutely hubris and so arrogant to think that we can survive without everything else. We come from this planet" (Kolbert 2021, p. 109). In short, a certain respect for the power of natural processes, and the magic that inheres in corals themselves, persists even in the midst of high-tech interventions. The attitudes expressed by these scientists suggest that a willingness to consider hi-tech interventions, genetic and otherwise, can go hand in hand with a humble attitude of world-repairing.

Gates and her frequent collaborator Madeleine van Oppen (who is exploring assisted evolution with algal symbionts) have routinely denied that their work constitutes genetic engineering. They emphasize that their techniques hew to "old-fashioned" selective or cross-breeding, or that they are simply doing what evolution would do, but faster. And it is true, to some extent, that super corals happen in nature; bleaching disasters reveal the "winners"—the survivors of a kind of accidental assisted evolution, triggered by human activity. Still, Gates insists, "We're not creating anything new: we're doing what nature does, and just trying to find ways to do it more quickly" (Braverman 2018, p. 236). The claim that assisted evolution creates nothing "new" can be debated, as I have suggested. But more significant than whether a certain technology creates novelty is how researchers understand and express their motivations and their relationship to broader contexts in which the technology is used. What matters is the deference they show to sustaining, as best we can, processes that did not originate with us and are not fully at our disposal. We might think of Gates and van Oppen as reluctant designers. Braverman, for example, discerns in the attitudes of these two scientists, and those in the broader coral community, a certain amount of "trepidation" about intervening, and a preference for minimal intervention whenever possible (Braverman, p. 218). The danger of humans assuming the designer role in assisting evolution is that doing so might foster or reinforce dispositions, and ways of understanding what it means to be human, that are injurious to nature (and possibly to ourselves)—and which likely caused harm to nature in the first place (Filbee-Dexter and Smajdor 2019). To be sure, these worrisome attitudes and dispositions do exist among genetic engineers and synthetic biologists. They are vividly encountered, I believe, in de-extinction initiatives. Turning to the moral and religious imaginaries of those who tinker—without trepidation—with technologies surrounding de-extinction (and intentional extinction) brings into sharper relief a form of enchantment with fabrication and control of life that is genuinely troubling in its refusal of relational, communitarian values, and networked knowledge and action. In de-extinction, the magic of salvific technologies and the wonder of human genius trump the magic and mystery of encountering other complex beings and forms of agency. The rituals of laboratory experimentation and rationality are seen to preclude the need for rituals of mourning, and myths of Darwinian competition set a confident masculine tone.

## 7. De-Extinction and Denial

In some of the most widely publicized and celebrated projects aimed at de-extinction, genetic technologies are mobilized with striking disregard for ecological and social contexts, and the key forms of relationality that shape organisms and species into what they are.

Organisms, in the imaginary of some aspiring de-extinctionists, are machines comprised of parts to be interchanged, omitted, or enhanced. Genes are deployed as usefully "selfish" units of matter that can be programmed at the engineer's bidding, even in the service of excising unwanted parts of an ecosystem altogether—a project that may culminate in programmed extinction. For these scientists, de-extinction and extinction are both attractive possibilities, menu options on a list of interventions from which we select. Especially noteworthy is the prominence of a neo-Darwinian, gene-centered vision among pioneering researchers working with technologies of de-extinction (and relatedly, extinction), in contrast to moral imaginaries that inspire and are inspired by work with corals. De-extinction projects reflect the instrumentalist mindset and individualistic, anthropocentric imaginaries of the researchers and entrepreneurs leading the charge. Enchantment with machines and organic life as units of information or "code" contrasts in interesting ways with the celebration of embodied, distributed knowledge and agency seen among many admirers of coral communities. In the cases under examination here—focusing particularly on entrepreneur, futurist, and ecopragmatist Stewart Brand, and work conducted under the auspices of George Church's Harvard lab—humans are positioned as godlike sculptors of life, conferring meaning and value to otherwise deficient, meaningless, or even immoral natural processes.

Does the comparison already sound a bit overdrawn? To be sure, the critique of godlike scientists has become cliché, and in the context of de-extinction, the charge is often issued in tandem with the usual tiresome references to *Jurassic Park*, Frankenstein, and other narratives that function as a shorthand for forbidden knowledge. Leveling the charge of playing God, moreover, often acts as a diversion from more productive discussions of how humans ought to relate to nature and what we might call divinity. That said, however, the attitudes displayed by prominent de-extinction advocates do exhibit an investment in playing God, if by playing God we mean something like asserting mastery over nature and reveling in unlimited possibilities of creating and remaking life. For some in the de-extinctionist camp, the act of making and remaking life is pursued as if nature were a storehouse of potentially (but not yet) valuable materials that can be swapped out, interchanged, and replicated as needed to achieve creative fulfillment for the designer, and in ways only tangentially related to environmental concerns or objectives. High-minded humanitarian and environmental rationales for this work often appear strategic and ad hoc (Sideris 2024).

## 8. The Tool Imaginary

A particular vision of creativity and fulfillment has roots in some of Stewart Brand's earliest and most widely cited oracular pronouncements. "We *are* as gods", Brand announced in the inaugural issue of the *Whole Earth Catalog* in 1968, a do-it-yourself manual for a tech-savvy counterculture. Brand's early articulation of this credo was followed by the words "and we might as well get used to it". Brand, who is 84 years old, has tweaked this dogma over the decades but never disavowed it. To wit: "We are as gods and we might as well get *good* at it" and, later, "We are as gods and we *have* to get good at it".[16]

But what did Brand mean by aligning humans with gods in the first place? What did he imagine gods to do? For Brand, getting good at (or used to) being God was, and is, largely an individual, independent pursuit. Gods, in Brand's vision, are autodidactic, self-created creators. Brand's initial elaboration in the *Whole Earth Catalog* of what it means to be god celebrates (in his words) the "power of the individual to conduct his own education, find his own inspiration, shape his own environment . . ." (Whole Earth Catalog 1968, p. 2). The catalog was conceived as a toolkit for expressing individualist urges and excellences. Brand is considered by some to be the father of online networking.[17] Yet, for all his ties to systems and systems thinking, networks, and networking, his vision of godlike creativity was borne not of a relational impulse, but of the quest to reinvent oneself and the world. Brand celebrates direct power that "eschew[s] institutions in favor of individual empowerment" (Weiner 2018). The ranks of his devotees include ardent cyber-optimists, tech visionaries,

entrepreneurs, and others enamored of "tools" and "tool talk", ranging from Steve Jobs to the founders of Facebook, Stripe, and Airbnb. For those in this lineage, tool talk "encodes an entire attitude to politics", specifically, a *rejection* of politics, in favor of clever "tinkering". Tools are seen to enable networks and communities; they are not constrained by them.

In the imaginaries of Brand and many who came after, countercultural proclivities mingle promiscuously with libertarian values and entrepreneurial excess, a peculiar mix of commitments sometimes labeled the "Californian Ideology" (Barbrook and Cameron 1995). Brand nowadays considers himself post-libertarian, but his apolitical, acontextual visions of society live on in his projects. Brand considers ecopragmatism a reasoned-governed approach, in contrast to the emotional intensity, romanticism, despair, and melodrama of the traditional environmentalism he abjures.[18] He presents ecopragmatism as an alternative to ideology. De-extinction is Exhibit A of Brand's tool mentality—and of the suspect forms of magic and enchantment that define his (and his collaborator, Church's) style of apolitical play with the basic elements of life.

In 2012, Brand and his wife Ryan Phelan co-founded a nonprofit organization called Revive & Restore, with a stated mission of "genetic rescue" and biodiversity enhancement for endangered and extinct species. A centerpiece of that initiative was the aforementioned wooly mammoth project. Mammoth revival is proffered as a form of climate mitigation, a massive rewilding experiment to slow the melting of arctic permafrost and the attendant release of methane, a potent greenhouse gas. Revive & Restore hopes to populate arctic regions with herds of lab-bred mammoth-like creatures whose trampling and grazing behavior might recreate the steppe ecosystem that existed in the Pleistocene, when megafauna dominated. Ecological mechanisms of the steppe ecosystem, including the predicted behavior of reintroduced mammophants, would have a cooling effect on the climate and, it is hoped, keep permafrost frozen and methane contained.

Brand and Phelan have connections with the father and son team, Sergey and Nikita Zimov, who own land in Northern Siberia dubbed Pleistocene Park, where the rewilding experiment would unfold. For genome engineering expertise, they have turned to George Church whose talents lie in the creation of novel DNA sequences. Church's overarching ambition is to rewrite the genetic code (Nair 2012). The mammoth project is just one of many adventures in genetic technology that have defined his life and career. Magical encounters with machines and machine-like entities set him on his course. An early fascination with computer technology (before the advent of widespread personal computers) shaped Church's lifelong penchant for viewing life as bits of information to be decoded, rewritten, and repurposed. In 2009, he developed automated genome engineering methods that helped pave the way for the creation of the first living cells from man-made instructions— life in a lab, more or less, or as fellow biologist and wealthy entrepreneur Craig Venter proudly proclaims it, the "first self-replicating cell on the planet to have a computer for a parent" (Biello 2010).

Church's breakthroughs in genome sequencing have also led to the proliferation of private enterprises like 23 and Me and Navigenics, a "personal genomics" company. He has long been fascinated by the prospect of de-extincting mammoths. Ideally, for Brand and Church, the newly created organism, a de-extincted mammophant, would be gestated and raised by an elephant mother, but this is problematic for many reasons. Elephants have distinct characteristics and behaviors, so a mammoth-like creature raised by an elephant might not exhibit the mammoth behaviors needed to make the rewilding/climate project a success. Additionally, as noted previously, Asian elephants are endangered, so recruiting them for surrogacy is inadvisable. Church and his team therefore hope to fabricate artificial wombs, manmade contraptions in which the fetus would gestate for the unusually protracted elephantine period of approximately two years, reaching a birth weight of close to 200 pounds. Once birthed, the creature would somehow be coaxed into behaving like a mammoth.

### 9. Impossible Worlds of Nonrelation

One can discern a through-line from Church's pursuit of synthetic, lab-created life and personal genomics, to machine-gestated hybrid creatures who carry undead genes. The mammoth revival project is billed as a nature-based solution to climate change. Yet the motivation for the project bears little obvious connection to the values that animate environmental conservation and restoration. From mammoth conception to (possible) mammoth introduction, the whole endeavor announces its refusal of relationality and of the *storied* nature of organisms and ecosystems. Consider Church's vision of gestating a lab-created fetus in a mechanical womb that treats uterine environments as interchangeable things, manufactured objects that exist in the world apart from the bodies they belong to, or the creatures that grow within them. In reality, of course, the uterine environment itself affects how genes are expressed in the gestating creature. A mother's hormones trigger developmental changes in fetal life, governing when and how certain genes are expressed. However closely related to mammoths, surrogate elephants cannot replicate the (long-extinct) mammoth's uterine environment.

Assuming that the gestational phase of the project can get off the ground, additional, and seemingly insurmountable obstacles remain, many of which are social and relational. A few facts about elephant maturation and social arrangements will suffice to illustrate the unattainable vision—and immoral imaginary—of mammoth de-extinction. Elephants and, we might assume, mammoths, are intensely social creatures. They live in tight-knit family herds that often contain multiple generations. These groups consist of related females who share complex relationships with one another as they work together to raise their young. Females take on dominant roles, and some elephant species have clear matriarchs. Asian elephants reach sexual maturity at 8–13 years of age, but females often do not reproduce until they are closer to 16 or 17 years of age, at which time they usually give birth to only one baby. Mothers nurse for two to four years, sometimes longer. Many males of the species do not mate until the age of 30. Like many intelligent creatures, elephants are long-lived, slow developers. These features (and others) define the bare minimum of what it means to be an elephant. They speak to qualities that do not attach to individuals only, much less to genes, but rather play out in relationships among organisms and between organisms and their environment. While it might be possible to edit in (or edit out) genes linked to certain behaviors—for example, matriarchy—the trait means nothing in the absence of social and ecological contexts and dynamics. The challenges these features present for anyone hoping to engineer herds of such creatures are daunting at best. Needless to say, we do not even know what additional challenges a mammophant might bring, since we know so little about mammoths.

Church and his team envision lab-bred creatures raised by humans. They estimate the number of mammoths needed to recreate the steppe ecosystem and arrest melting permafrost to be around 80,000 (Wray 2017). Read that sentence again. And imagine for a moment the task of orchestrating and synchronizing all of the complex interactions that allow these creatures to successfully gestate, develop, mate, give birth, form functional social arrangements, learn from one another, and navigate complex environments—and then imagine recreating those dynamics to generate a herd of 80,000 animals, one captive-raised and captive-bred animal at a time.

What makes it possible for enthusiasts of mammoth revival even to conceive such schemes is a complete disavowal of organisms and species as constituted by ecological and social relations. Colossal, the company that has bankrolled the mammoth project since 2021 (co-founded by Church and serial entrepreneur Ben Lamm), runs a slick website featuring a menu of "disruptive" conservation technologies. The site downplays the barriers to realizing mammoth de-extinction in brief step-by-step descriptions that are almost laughable in their casual oversimplification. One reads, for example: "Help with nutrition and social interaction for young calves to thrive." The yawning temporal gap that separates extinct mammoths from current ecosystems is greatly minimized. Mammoths, the site informs us, went extinct "only 4000 short years ago" (Colossal.com n.d.). In geological

time, 4000 years may seem insignificant, but it is more than enough time for an ecosystem to move on from the demise even of a keystone species.[19] Colossal meanwhile plays up the increasingly untenable theory that humans hunted mammoths to extinction—the so-called overkill theory—thus giving greater weight to mammoth extinction as unnatural and their resurrection as the just and proper course (in fact, most scientists now believe mammoth extinction was brought about—ironically—by climate change). When pressed on the sketchy details of mammoth resurrection, Church quickly changes tactics from promising the imminent return of a mammoth to claiming that the public has misunderstood him. He and his team are "really resurrecting genes, not species", he insists, as if genes have inherent value apart from organisms, and ecological and evolutionary processes (Amanpour & Co. Public Broadcasting System 2019).

Let us assume that the technology can be perfected to the point that a resurrected organism is an authentic copy of its extinct counterpart. It still makes little sense to claim that a *species* has ceased to be extinct when an individual has been brought back. Species are dynamic, aggregate entities, living repositories, with long evolutionary histories shaped by complex interrelationships with other creatures and their natural environment (Sideris 2024). These relationships, which extinction erases, are precisely what de-extinction fails to recover: "Engineered reproductions" of organisms, as philosopher Ben Minteer writes, "will not have evolved in relationship with other species and within a given ecological setting over millennia" (Minteer 2018, p. 111). Natural histories and ecological relationships lie at the root of what makes species valuable, and their loss through extinction lamentable (Sideris 2024). There are many reasons to value the natural history of an unengineered creature, but an important one might be easily overlooked: a species' history and relationality "encourages the adoption of an attitude of humility toward them", Minteer argues (111). Perhaps, as he proposes, the root of human-caused extinction is a "self-regarding worldview" that feeds fantasies of mastery and fabrication. With de-extinction, the awe and sublimity once directed at nature is now directed at our own techno-prowess in manipulating life. The self-regarding worldview is consistent with the bankrupt imaginary of de-extinction that treats living creatures and systems as an extension of the individual's creative impulse to design and redesign life at the tinkerer's will. The peculiar magic that inheres in these creative endeavors is plainly discernible in de-extinctionists' understanding of life as a machine.

## 10. Machine Magic

To burnish their project's ecological credentials, Brand and Church promote mammoth resurrection under the heading of ecosystem recovery (Wray 2017). They concede (at times) that the organism created through these technologies is not a precise replica, but they argue that a decent proxy can plug a hole in an ecosystem left by the departed species. This argument fits with their broader understanding of life as consisting of interchangeable units. Brand argues that so long as there is an abundance of different species in an ecosystem, extinction itself, even human-caused extinction, is of no great significance (a claim that seems at odds with his preoccupation with mammoth resurrection). Extinction, he maintains, creates new opportunities for diversity to flourish. Organisms threatened with climate change can simply move somewhere else and hybridize, he suggests (Wray 2017, p. 69). The Anthropocene is "creative" in that climate change "tends to open the way for more species rather than fewer", resulting in a natural world that "as a whole is exactly as robust as it ever was" (Brand 2015). New parts can simply replace the parts that have vanished. The pieces can be reshuffled.

Church and Brand hold sympatico views of life. Church defends an account of animal life that is so crassly and anachronistically Cartesian that it seems to come from the mouth of a movie villain. "All organisms", he announces, "are mechanical in the sense that they're made up of moving parts that inter-digitate like gears. ... They are atomically precise machines" (Der Spiegel 2013). Church's faithful recitation of the animal-machine theory occurs in the course of a wide-ranging and frankly bizarre interview in the German

publication *Der Spiegel*. The interviewer, listening as Church blithely outlines potential future projects, ranging from the prospect of de-extincting Neanderthals to abandoning Earth for other planets, appears increasingly nonplussed by the great scientist's disregard for matters of ethics and propriety. Finally, this question is put to him in point blank fashion: "Mr. Church, do you believe in God?" Church utters a bromide about the power of faith in human history. The interviewer stops him short: "But you're talking about other people's faith. What about your own faith?" "I have faith" Church proclaims, "that science is a good thing". He explains that the word "awe" was practically invented for scientists. "A poet sees a flower and can go on and on about how beautiful the colors are", he offers pedantically. "But what the poet doesn't see is the xylem and the phloem and the pollen and the thousands of generations of breeding... All of that", Church concludes with a flourish, "is only available to the scientists" (Der Spiegel 2013). Assuming Church is in earnest (it is difficult to judge because his commentary appears parodic in its recourse to such well-worn tropes), his response provides a succinct summary of the secular enchantment of the de-extinctionist. Genuine awe—awe at the organism reduced to its component parts—is the purview of the scientist alone (Sideris 2017).

Church's animals-as-machines perspective is of a piece with a broader vision of life that tasks humans with decoding, copying, and engineering diversity from the raw materials at hand—a sacred obligation Church has dubbed "regenesis", the reinvention of nature and ourselves (Church and Regis 2012). Church's pragmatic, human-centered understanding of the value of diversity forms an interesting contrast with celebrations of natural diversity, like those issued in praise of coral reefs. Diversity for Church is a hedge or bulwark against human extinction specifically. The task of synthetic biology is essentially to utilize existing life as a template from which to generate more diversity. Resurrecting Neanderthals, he claims, is a form of "societal risk avoidance", as the main objective in de-extincting them is to "increase diversity" (Der Spiegel 2013) Neanderthals might prove useful, Church suggests, because they might think differently than "we" do—a benefit of diversity. Church has some very particular disaster scenarios in mind, for which Neanderthals might serve as insurance. "When the time comes to deal with an epidemic or getting off the planet or whatever", he predicts, "it's conceivable that their way of thinking could be beneficial" (Der Spiegel 2013).

Getting off the planet or whatever. In Church's apocalyptic fantasy, scientists synthesizing diversity in the lab will be the ones to rescue humanity from destruction. Church heads up the Wyss Institute for Biologically Inspired Engineering. His lab has supervised and inspired projects ranging from age reversal to seeding other planets with life, to intentional eradication of pest species. The research projects of his mentees offer a glimpse into the culture and broader reach of the world-famous Church lab.

One such protégé is Kevin Esvelt, an avid supporter of gene drive technology to control or even eliminate "unwanted" species. Gene drives are self-propagating genetic elements that bias inheritance, spreading genomic alterations through a population very quickly, even if the genetic elements confer a disadvantage to the organism, like sterility.[20] Emerging techniques using CRISPR can force a particular edit to be inherited by all of an organism's offspring, efficiently driving a trait through an entire population.[21] Climate change has given urgency to some lines of gene drive research, because mosquito-borne diseases are on the rise, and are expected to continue their uptick in a changed climate. Hence, one of the main applications of gene drives is mosquito populations to control or eliminate disease vectors in malaria-carrying mosquitoes.

## 11. Selfish Scenarios

Gene drives are depicted as "selfish genes" par excellence, a term popularized by Richard Dawkins who understood evolution to take place at the genetic (not the organismal, species, or population) level, as individual genes battle it out for survival. On this account, what looks like cooperation or altruism at the level of organisms is in fact driven by selfish genes striving to propagate themselves, beyond even the death of the individual who carries

them. Because genes are shared among close relatives, the fate of a given individual matters less (from the gene's "perspective") than the survival of the genes in *any* body.[22] Central to much of selfish gene theory are the intertwined ideas of evolution as competitive strife and genes that "program" their carriers. A great deal of mythic baggage has accumulated around selfish genes. "We are survival machines—robot vehicles blindly programmed to preserve the selfish molecules known as genes", in Dawkins' famous diagnosis. "We, and all other animals, are machines created by our genes" (Dawkins 1976, p. xxi, 2). Over the decades, selfish gene theory has been challenged, amended, contextualized and, among some biologists, rejected outright. One area of biology (or biotech) where the concept still holds sway is in the discourse on gene drives.

For Esvelt, who christens his MIT lab "Sculpting Evolution", the appropriate ends toward which scientists must guide nature are determined by a utilitarian calculus in which humans reign supreme as creatures endowed with unsurpassed capacities for both intelligence and suffering.[23] Note that so powerful are manufactured gene drives that they might wipe out a species altogether (intentionally or otherwise). Esvelt, the gene-driver, acknowledges the toll human-induced extinctions have caused and are currently causing, but his brand of (oddly speciesist) utilitarianism posits humans as the most valuable species owing to our sophisticated capacities.[24] Creating more creatures like us would redeem the value destroyed through mass extinction events involving less impressive species. Thus, he reasons that "if we terraform Mars and seed it with life, that will more than outweigh any of our past sins" (Esvelt qtd. in Braverman 2017, p. 62).

It is easy to see how the Dawkinsian life-as-gene-machine imaginary aligns with Church's animals-as-interlocking-gears perspective, and vice versa, even if theories and technologies of the gene have grown more sophisticated since Dawkins' *Selfish Gene*. Esvelt has a particular axe to grind when it comes to the natural world. Like Church, he describes his mission in language reminiscent of a stock character from a bygone era, as he denounces nature as an abomination, red in tooth and claw. The natural world is an arena of unrelenting bloody strife and natural selection is "heinously immoral". Esvelt is determined to address this "fantastic degree of suffering", to redeem nature's intrinsic evil (Specter 2017, p. 36). As a child, he was captivated by Michael Crichton's book *Jurassic Park* (Specter 2017, p. 36). But what he terms his "real conversion" to biotech occurred with a childhood trip to the Galapagos. He became "fascinated" with the idea that complex systems—organisms—were all "written in the language of DNA". At that point, he knew. "I wanted to spend the rest of my life learning how to rewrite the genes of organisms to make some extremely useful and interesting things" (Specter 2017, p. 36). In these magical childhood moments, and the narratives of conversion that replay them, lie the seeds of the adult scientist's religious repertoire. Like Church, Esvelt believes that features inherent in nature itself cry out for a human upgrade. Nature exists to be rewritten, to make interesting things. It is troubling, but perhaps not surprising, that—as Braverman observed in interviews with Esvelt and other gene drive scientists—all were "undereducated" in the finer points of ecology. She concludes (with admirable restraint) that "their views about nature-human-animal relationships could benefit from some sophistication and historical contextualization" (Braverman 2017, p. 71). One might say that Esvelt has no view of nature-human-animal *relationships* at all.

In her critiques of the selfish gene concept and evolutionary mythmaking more generally, the philosopher Mary Midgley argued that the "selfish" descriptor came not from science but from "a fresh outcropping of the strong, egoistic, individualistic strain in our political and moral thinking" (Midgley 2001, p. 196). The concomitant view of nature as violent and immoral also owes much to popular mythmaking around Darwinian theory. Scientists and philosophers who assail nature as immoral, evil, or absurdly meaningless often dismiss those with more positive views of nature as starry-eyed romantics. Esvelt, for example, mocks the idea that "nature is the essence of goodness" (Specter 2017, p. 207). But as Midgley understood, the same thinkers fail to recognize their *own* affective investments and the degree of mythmaking that goes into pronouncing the world cruel and senseless

(Midgley 1985). Jettisoning the "romantic" vision does not excise the emotion and drama; it merely replaces one evolutionary drama with another— nature red in tooth and claw. These moral imaginaries and mythic renderings of our place in nature can easily become entangled in the practice of science; they appear, for those in their thrall, to speak truths about nature that are equivalent to science. If ethical decisions are rendered according to a crude utilitarian calculus; if nature is cruel, immoral, dysfunctional, and organisms are machines we can decode, there is surely little reason to refrain from gene-driving a species like mosquitoes to extinction if it benefits mankind in the struggle to survive climate change. The genetic engineer becomes the savior who intercedes on our behalf, the hero with the techno-fix, who stands between us and the apocalypse.[25]

## 12. Navigating World-Pictures

Midgley's claims about evolution *as* religion, and the moral and affective investments of scientists who deny the existence of "feeling" in regard to nature, deserve another look as we draw this lengthy analysis to a close. Like coral scientists who find hope in genetic technologies, de-extinctionists (and those, like Esvelt, whom we might call extinctionists) also see these technologies as bringing a much-needed good news story to a world drowning in dire headlines. These genomic feats inaugurate a shift away from what Brand dismisses as the "constant whining and guilt-tripping" that has dominated the conservation community, toward a celebratory mood of "high fives and new excitement" (Brand 2014). Brand and others in his cohort present Revive & Restore as a counterpoint to the mournful passivity that plagues traditional conservationists. To grieve is to do nothing. "Don't mourn, organize!" Brand advises (stealing a line from labor activist and songwriter Joe Hill). Focusing on extinction introduces an unnecessary "emotional charge", he believes.

The claim that de-extinction, and other "active" genetic interventions pursued by Revive & Restore, replace emoting with *doing* something is belied by Brand's own effusive, highly emotive discourse. Brand depicts de-extinction as a "wild scheme" that "could be fun" (Qtd in Rich 2014). Whereas Esvelt upbraids romantics who see purity and goodness in nature, Brand and Phelan seek to distance themselves from negative feelings of hopelessness or guilt that they associate with traditional environmentalists—those whom Brand labels "lazy romantics" (Brand 2015). De-extinctionists, eschewing romance, want to claim the mantle of rational planetary management. Ecopragmatism, understood by its proponents as the triumph of action over navel-gazing self-recrimination and despair, is the guiding philosophy of Revive & Restore. And yet, the focus on bringing back ultra-charismatic creatures like the mammoth is very much about how these creations would make us *feel*. The fact that humans might accomplish these astonishing feats—the idea that we are indeed "as gods"—speaks volumes of the intensity of emotion and excitement that drives de-extinction. Unlike the coral scientists who acknowledge emotional investments, these species revivalists have failed to grasp the profoundly emotional commitments that fuel their own imaginaries and their longing to bring into existence highly improbable worlds.

The emotional landscape of these de-extinctionists (and potential extinctionists) is hiding in plain sight. But it is an impoverished landscape, lacking gravitas. What is missing is the *relational* landscape that might allow these aspiring creators of new, unlikely worlds to place what they call "hope" into an appropriate moral, social, and relational context. Missing too is what Braverman and others call active hope—the sort of hope that has confronted genuine grief and despair, dwelt with them again and again, and emerged refined and reoriented by them (Sideris 2020). As Donna Haraway has observed, there is something decidedly puerile about the imaginaries of those she calls "Stewart Brand types", who exhibit an "incapacity to mourn . . . to be finite", and are therefore forever grasping for new tools without comprehending the losses they seek to "fix". They cannot understand that death and loss are real, and thus they appear oddly "blissed out" by their own unscathed and intact privilege. They "have no idea what their own positionality in

the world really is", Haraway astutely observes. In an apt characterization of the magic that pervades their high-tech, individualist adventures, she discerns in these tinkerers "an almost Peter Pan quality". They never grow up (Weigel 2019).

Indeed, contrasts between communal coral imaginaries and the apolitical Promethean dreams of de-extinctionists emerge so vividly in a side-by-side analysis that the comparisons almost write themselves. The contending images are archetypal, familiar, and hard to resist. On the one hand, the sacred holobiont, a symbol of relational life and collective labor. On the other, the apotheosis of the selfish gene locked in eternal zero-sum strife. The life-giving coral, synonymous with fertility, vitality, blood, and birth; the bloodless artificial womb of the de-extinctionist, and the sterility of the self-destructing gene-drive organism. The irreducibility of the coral superbeing; the lab specimen of the reductionist engineer. The coral as designer and architect, the originator of life; the human genetic engineer as life's intelligent designer. The coral world-maker. The human world-maker. Above all, perhaps, the ethos of the nurturing, networked female scientist versus the ego-driven competitiveness of her masculine counterpart. These juxtaposed images and mythic motifs are, of course, overdrawn. They traffic in well-worn, recurrent narratives, stereotypes (gendered and otherwise), and stock imagery. The binaries are too simplistic. But they contain important truths nevertheless. As Midgley understood, the point is not to extricate ourselves from all world-pictures, for that is not possible. We must instead navigate intelligently among our imaginative visions and our religious repertoires, and choose which ones to live by. We need to interrogate our myths and scrutinize our enchantments, not banish them outright. The question is: which of these socio-ecological imaginaries do we wish to inhabit as we move into a phase of intensified technological intervention in life processes? Which visions should we lock in and which should we lock out, if we hope to live well together in climate-changed future?

**Funding:** This research received no external funding.

**Institutional Review Board Statement:** Not applicable.

**Informed Consent Statement:** Not applicable.

**Conflicts of Interest:** The author declares no conflict of interest.

## Notes

[1] One could just as easily call them, as Beaman and Stacey do, "nonreligious" imaginaries. Nonreligious is not necessarily the same as an atheistic stance; rather, nonreligious describes individuals for whom *not* being religious is *not* significant to their identity. For the sake of consistency across terms and with the concept of religious repertoires, I will call them religious while acknowledging the inexactitude and myriad connotations of the word.

[2] Rather than label these tactics with more traditional terms like conservation or restoration, I choose to label them for the time being as management. Whether or not they constitute restoration or conservation depends on the details and is not easily determined in advance. Whatever else is going on, organisms and species are being managed in relation to goals that may or may qualify as either restoring or conserving, and may in fact have no robust "environmental" justification at all, as is the case with some examples of what is called de-extinction.

[3] For example, Ruth Gates, a beloved coral scientist who died in 2018 is featured prominently in the film *Chasing Coral* (2017). Harvard geneticist George Church is featured in a few documentaries, including one focused largely on the life and legacy of techno-entrepreneur and ecomodernist Stewart Brand titled *We Are as Gods* (2020).

[4] However, the distinctly *non-relational* nature of de-extinction is not simply a function of the impossibility of forming meaningful attachments to extinct creatures (though that is part of it). Denial of relationality is in some sense built into the very ideology of de-extinction, as I will argue.

[5] Cloning techniques come closer but are of little use with extinct creatures for whom there is often no intact DNA.

[6] Indeed, Braverman's study underscores the diversity and disagreements within the coral community.

[7] Questions of human intervention in the natural environment are not new and have long been at the center of environmental ethics. The key question in much of environmental ethics, since its inception as a field of study in the 1970s, has not been whether humans should *ever* intervene, but rather when, how, why, and, if so, to what extent? Contemporary ecomodernists or ecopragmatists who define themselves against those they consider "traditional" environmentalists often set up a strawman account of what that tradition values and practices. Despite what these boosters of a human-managed planet might claim,

traditional environmental conservation and restoration has never defined itself by a hands-off approach to some purportedly pristine and pure entity called nature. At the same time, the fact that humans have long intervened in nature should not be read as an "anything goes" mandate to remake the planet, simply because it has already been "used", as ecomodernists are fond of saying. (See for example Erle C. Ellis, "Ecology in an Anthropogenic Biosphere." *Ecological Monographs*, 85(3), 2015, pp. 287–331).

[8]   Scientists categorize bleaching events according to their severity and scale: local, mass, and global.

[9]   Indeed, I remain perplexed as to whether to use the word coral or corals. Navakas (2023a) uses the plural, Braverman (2018) the singular.

[10]   See for example Juli Berwald's (2022) frequent encomia to corals.

[11]   See Gilles Deleuze and Félix Guattari, *A Thousand Plateaus: Capitalism and Schizophrenia* (University of Minnesota Press, 1980).

[12]   An example of how intimate observation of lifeforms in the oceans can foster empathic values and novel insights regarding radical otherness is seen in the widely acclaimed (and critiqued) film *My Octopus Teacher*.

[13]   Wildlife artist Bob Hines who worked alongside Rachel Carson during the writing of *The Sea Around Us* reported that regardless of the late hour or how exhausted they were, Carson insisted on returning sea creatures to the exact spot from which they were taken. (See "The American Experience: Rachel Carson's Silent Spring". Public Broadcasting System, 8 February 1993).

[14]   And yet, we should refrain from uncritically romanticizing collective labor as symbolized by corals. As Navakas argues, celebrations of communal labor that draw on coral imagery sometimes *erase* the lives of exploited and enslaved workers whose bodies are absorbed and forgotten in the process of "growth". A darker vision of corals suggests that "generation after generation, from birth until death without leaving, the workers build a structure that excludes them . . . Meanwhile the reef rises from their laboring bodies which endlessly merge to become a coral island that supports those who did not produce it and do not remember who did" (Navakas 2023a, p. 3).

[15]   Some scholars, such as Sophia Roosth, view the crochet reef in a more pernicious light, aligning the techniques and motives of its practitioners to the objectifying aims of synthetic biology—an obsessive desire to control and manipulate life. I do not find this critique particularly persuasive; there is nothing obviously "post-organismic", as she claims, about the reef project. In fact, the problematic imaginary of life-fabrication she discerns has much more in common with de-extinction experiments and related projects. (See Roosth 2013).

[16]   For an example of the third iteration of his famous maxim, see Brand, "We are as Gods and Have to Get Good at It." *Edge* (2009). 18 August 2009. Brand seems to have misquoted himself or misremembered the original phrasing in the *Whole Earth Catalog*. He recalls his own credo as having said "might as well get good at it", whereas the 1968 publication of *Whole Earth Catalog* actually reads "might as well get used to it".

[17]   See Fred Turner, *From Counterculture to Cyberculture: Stewart Brand, the Whole Earth Network, and the Rise of Digital Utopianism* (University of Chicago Press, 2006).

[18]   Ecopragmatism, also known as ecomodernism, embraces geoengineering, biotechnology, genetic engineering, nuclear energy, intensified urbanization, and other technological controls as consistent with environmentalism. (See Brand, *Whole Earth Discipline: An Ecopragmatist Manifesto*. New York: Viking, 2009).

[19]   Very recent research suggests that in fact the extinction date of 4000 years ago may be skewed by ancient DNA samples by as much as thousands of years, meaning that mammoths actually went extinct much earlier. (See Bas Den Hond 2022). Again, there is much we don't know.

[20]   Gene drives do exist in nature (some confer no fitness advantage) but their power is constrained by evolutionary pathways.

[21]   Given that some coral scientists want to create corals that are resistant to warming oceans, gene drives seem like one possible way to achieve those ends. Some scientists (though not Gates or van Oppen) have explored this avenue with corals, though the possibilities are limited in the near term.

[22]   Dawkins often insists that he never meant to imply that selfish genes make for selfish individuals, but his book is full of statements contradicting that claim, such as the idea that we must consciously build a cooperative society because humans are "born selfish".

[23]   Esvelt subscribes to a philosophy called "effective altruism" that is increasingly common among world-shaping billionaires and technocrats. Some of his endeavors, under the auspices of his company BioSecure, received funding from the disgraced cryptocurrency mogul Sam Bankman-Fried, a fellow practitioner of effective altruism. https://www.science.org/content/article/crypto-company-s-collapse-strands-scientists (accessed on 2 November 2023).

[24]   *Oddly* speciesist because Esvelt and other "effective altruists" look to utilitarian philosopher Peter Singer, who sought to dismantle speciesist biases that grant special moral consideration to humans and human suffering, over against other lifeforms.

[25]   Esvelt positions himself as a champion of open science, continually warning the public about the kinds of technologies he himself is unleashing. He is fond of quoting J. Robert Oppenheimer who did the same (only after his creations were unleashed), and others have made the same connection. "Not since Robert Oppenheimer has a scientist worked so hard against the proliferation of his own creation", one profile of Esvelt reads (See Love (2019). Also see Rowan Jacobsen, "Deleting a Species", *Pacific Standard*, 7 September 2018. Available online: https://psmag.com/magazine/deleting-a-species-genetically-engineering-an-extinction (accessed on 2 November 2023).

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
