# Peer review of "Living Well Together in a Climate-Changed Future: Religious Imaginaries on the Cutting Edge of Genetic Technology"

_religions, doi:10.3390/rel14111426_

Round 1
Reviewer 1 Report
Comments and Suggestions for Authors
Overall thoughts:
This a beautifully written article that represents a crucial contribution, both to RS and to STS and transitions studies. It shows how some of RS’s core categories are crucial in understanding approaches to global problems. It will allow STS and transitions scholars to deepen their engagement with the imaginaries of those they research. It is moreover a profoundly insightful and well-researched article, full of rich reflections that can help us to better understand (non)religious repertoires and the underlying (and often dangerous) assumptions of those designing socio-ecological interventions. For these reasons, it is crucial that the piece gets published.
However, a re-working/ordering for the sake of clarity would make the piece much more accessible for readers, and thus significantly increase its potential for impact. See in particular my comments for p.11/23 and p.16/23. But please do read through all the comments quickly before addressing anything, as you may find that whereas some comments seem to dramatically stress the need for a grand re-construction, others suggest a path to a more subtle edit: perhaps just much more clearly signposting. See, for example, the comment for p.18/23 and the first suggestion offered in my comment for p.11/23
Comments written while reading:
From abstract: I am excited by the article and I feel it will offer an important contribution to the field. I felt a little work on syntax could have made it easier to understand that the two rival imaginaries mapped onto the two distinct groups. I also felt it would have been good to have some clarity as to what the author’s normative aim is – but that’s perhaps a question of style.
p. 2/23
2.1 I need to be sensitive here, which is why I have agreed to sign my name in the review. I’m not sure it’s my place to intervene in the interpretation of terms, and that is for the author to decide, but just a suggestion: You say that, “It is through imaginaries that practitioners of emerging technologies express their vision of life.” I would instead argue that an imaginary is something (albeit fluid) that we, in a sense, harbour (why I have elsewhere [Saving Liberalism, pp.81-4] talked of ethics as a constellation). This is probably a matter of changing one word (!) but I thought I’d share my two cents. Up to you.
2.2 I then think of an imaginary as partially made up of (and only partially because I would say, for example, that a syllogism like “all humans are mortal, Socrates is a human, therefore Socrates is mortal” is also part of my imaginary) religious repertoires: magic, myths, rituals, traditions.
By the way, I stick with religious repertoires, even for nonreligious imaginaries because, as I state in a forthcoming article:
“I maintain the religiously connotative prefixes because although words widely employed in the study of religion and nonreligion like “imaginary” and “worldview” are increasingly adopted in the religiously less literate social sciences, they tend to lose a lot of depth in the process.”
But I think it’s also a question of your expected audience. No real point here I guess – I’m just finding your writing useful to think along with.
2.3 I love the way you interlace magic as against-all-odds with magic as enchantment. I do this too. I feel there is a profound connection between the two which I haven’t yet felt able to articulate.
p.3/23
3.1 “religious visions”. It could be useful to have some consistency. You mention religion-like, quasi-religious, and now religious. I am happy to ignore because I feel I have a sense of your argument. Others may be more pernickety, as I have been in the past. For example, I tend to get my back up about the imposition of a world religions framework onto ideas and groups.
3.2 I find your style or writing beautiful. I think some would want firmer clarity about the concepts you are using and the paper’s claims and aims. I feel I get the thrust of what you’re exploring, and I hate the way the art of academic writing is often destroyed by the insistence on science like statements of hypotheses etc. However, I do feel it would benefit from being sharpened.
3.3 I like the point about the magic of both sides giving them celebrity status. It would be interesting to think whether this were an example of their magic turning into myth. Especially powerful if the documentaries themselves have drawn in new people.
p.4/23
4.1 “Given the lack clarity or consensus regarding terms like synthetic biology and genetic engineering, perhaps evaluations of their merit would do well to focus less on the tools involved than on the intentions, motivations, and imaginaries of the people using them” I love this point
pp.4-5/23 Your sketches of the two positions is wonderful. I think this can make an important contribution to RS, STS, and transitions lit simultaneously.
5.1 Paragraph beginning “emotional intensity” is interesting but it’s not clear to me what it adds. It’s actually quite distracting
5.2 Actually the whole section “An Orientation to Two Competing Imaginaries” could benefit from tidying. I see your argument as following the logic quoted in 4.1: there are lots of overlaps between the two when it comes to technology. The difference lies in the imaginaries. That could come across more clearly.
p.6/23
6.1 Concluding paragraph beginning “In evaluating” is excellent and crucial
6.2 The whole section “Coralation”, while being wonderful as a self-contained piece, nonetheless invites the question: how does this fit in the overall context of the argument. It would be useful if you could signpost that. I think what you’re doing is providing an historically and geographically broad reflection on the role that corals have played in the human imaginary, thus setting us up nicely for understanding both why corals are an important starting point for broader questions of the human imagination, and what is at stake in your analysis.
p.8/23 In the “Coral Magic” section (again beautiful on its own) it’s not immediately clear to me which side of the debate you are speaking to.
p.11/23 “Hopeful labor” This whole piece is full of excellent and engaging insights. I think that my biggest problem in engaging with it is in not understanding clearly enough what you are aiming to do (although I think it would be fair to say something like: Socio-ecological agendas and technological interventions are becoming so complexly intertwined that it is hard to assess their intentions, let alone their normative value. With this in mind, this article draws on concepts such as imaginaries and religious repertoires to unravel the ideals and assumptions that we find binding together with different ecological interventions); your strategy for achieving that aim (does each [cluster of] section[s] represent one imaginary, or does it offer a reflection on both [I feel like both could be correct]); or the argument of each individual section.
What I would suggest is one of three things:
1) If my hunch is right that the first half is about the one imaginary and the second the other, then just make this a bit clearer.
2) Let’s say you’re using a more firmly repertories based approach: have subsections on magic, myth, ritual, tradition (or whatever repertoires you deem important) and use these to differentiate the two imaginaries.
3) Or: keep the thematic section headings, but make it much clearer how each theme speaks to some kind of dividing line between the two imaginaries, and draw on repertoires as a means of differentiating them: eg. Magic of encounter with other beings vs magic of salvific technologies/magic of human genius (and then, in the middle perhaps[?], the magic of fiddling with evolutionary processes); myths of nature as collaborative vs. those of nature as competitive – or perhaps myths of nature as profound vs myths of human genius – or perhaps myths of embodiment vs myths of reason; rituals of underwater submersion, collective endeavour and embodiment vs. rituals of the laboratory experiment and rationality – or rituals of mourning vs. rituals of action
p.14/23
14.1 “De-extinction…” section is (obviously) much more clearly a discussion of one of the target imaginaries
14.2 “Does the comparison” paragraph is fantastic
p.16-23 As I read through the wonderful reflections on mammoths, I started to ask crucial questions for the piece: First, is this about corals, or mammoths? OR is it about rescue vs de-extinction more broadly? If that’s the case, it might be useful to set up the choice of cases at the beginning to prepare the reader. You’re actually talking about two, widespread forms of socio-ecological intervention, which often get mixed up together, and you are drawing on tools and critique from the study of religion to disentangle them.
p.18/23 As I get to half way through “machine magic”, I see that your core comparison is around rival uses of magic. I’d then also say that myths come in too, but that’s up to you.
The conclusion is perfect
Author Response
My comments below follow the chronology of the suggested edits the reviewer offered. I agree that the paper needed more signposting and tightening and even restructuring in some sections, and I have done a lot of that work. Specifically:
- I agree that the abstract needed reworking. I changed it significantly so that the rival imaginaries are easier to see and the whole overview of what the paper does is clearer.
- I reworded how I address the idea of imaginaries, hopefully more in keeping with how Stacey discusses them (e.g. as constellations and for those paragraphs that discuss imaginaries as well as the idea of religious repertoire, I reorganized much of that section so that it is clearer and I hope more in keeping with Stacey's account, although I may also be interpreting these concepts somewhat differently.
- I also eliminated my use of "religious-like" or "quasi-religious" and have just stuck with the terms "moral and religious imaginaries" without qualifying religious. I changed a footnote on this issue as well.
- I just cut the whole section on "emotional intensity" that the reviewer fond distracting. It really did not go there.
- I really reorganized the section around "An orientation to Two Competing Imaginaries and the section leading up to it on "Emerging Environmental Technologies" so that it clearly highlights the similarities and differences between de-extinction and coral restoration, listing the similarities first and then discussing how they diverge in terms of imaginaries, with the concluding idea that we should look less to the technologies themselves than in the motivations and affect and imaginaries in which they are deployed. I combined some sections as well so it was not so scattered.
- For the section on corals ("Coralation") I cut some of this out and also tried to indicate that what I'm setting up there is how corals have been part of the human imaginary. I hope the signposting and clean-up there helps.
- I cut a lot out of the "Hopeful labor" section, which the other reviewer thought also went on too long (on the Coral Crochet Reef). My aim there was to provide a concrete example of how collaborative visions, female leadership, and the model of the coral itself overlap in this project, and to set the crochet project as a foil for the individualist, Darwinian de-extinction and gene-drive projects discussed in the second half of the paper, to make those comparisons more vivid and tangible.
- I found the points about how to make clear the dividing lines between particular types of repertoires (rituals, myths, magic, etc) very helpful. Instead of completely changing the subheadings, I instead inserted comparisons about different kinds of rituals and myths in those sections to make it clearer what the competing repertoires are. I hope that is clearer. In some cases I even borrowed directly from your suggested phrasing to make my points clearer (see e.g. the passage that transitions from the discussion of corals to de-extinction (just before the section "De-extinction and Denial")
- It's a great question whether the essay is about, at some level, corals vs. mammoths. I ended up adding a sentence along the lines of the suggested phrasing about two widespread forms of socio-ecological intervention and using the tools of religion to disentangle them. Thanks.
I'm attaching a marked-up version of the essay so the changes will be visible. I will upload a clean copy elsewhere.
Reviewer 2 Report
Comments and Suggestions for Authors
Thank you for the opportunity to read and comment on this paper. I greatly enjoyed reading the work, and think that with some minor revisions, this paper will make an excellent contribution to the special issue on Religion and Climate Change. Overall, the paper is very well written and accessible to the reader and the material presented therein will be an important contribution to this volume and to the field.
I have a few substantive comments, and then a few more minor suggestions for the author as they work to revise and finalize this paper.
Foremost, it would be benefial if the author could draw out the argument that "these 'construals' of oneself in relation to others and the wider world (both social and ecological) constitute religious or religious-like imaginaries" throughout the paper. The through line gets a little lost in some of the later paragraphs especially, so if the author could reference back to these "imaginaries" every so often, I think this would be helpful for the reader.
The section on the coral crochet group seems to be quite long and descriptive, and though I think it is an important example, I'm not sure how useful or necessary the long description is for the purposes of this paper.
I also think that some reorganization and/or clarification would be beneficial in the section An Orientation to Two Competing Imaginaries - it gets a little confusing for the reader what the "two competing worldviews" are, but I think some simple moving around of the sentences will help clarify that. Later in that paragraph, I think it might also be interesting for the author to think about relationality in terms of the "genetic gap" between extinct species and those that could be "brought back", which might enrich the discussion about living "in relation". How, for instance, does Ben Minteer's observation that de-extincted species would be "arriving with suspiciously blank passports" factor into the conversation about interrelationships (evolutionarily and not) with today's species.
In the section on Hopeful Labor the 10th full paragraph in that section seems to be a bit out of place. I wonder if it might not be better to move this earlier in the paper, or to trim it altogether.
In terms of small things - there are some instances of word/phrase repetition back to back in the paper, and then there are also a few sentences that seem to be missing words or phrases. I think a careful proofread will help the author catch most of these, so it's just something to be aware of.
Again, thank you for the opportunity and I look forward to seeing (and using the paper in my classroom) once it is published!
Author Response
I'm attaching a marked up copy of the essay so the (fairly dramatic!) changes can be seen.
Regarding this reviewer's specific suggestions, I did the following:
I included much more signposting to make more of a through-line of the religious framing of the essay (religious imaginaries, religious repertoires, myth, magic, ritual, etc). Both reviewers suggested this.
I cut out a considerable amount of various sections and combined others. I cut a lot out of the section on the Crochet Coral Reef project which was far too long, and put some of it in a footnote. I kept the rest because I felt it provides a good foil for talking about the very individualistic, neo-Darwinian dimensions of de-extinction and gene drives.
Both reviewers felt that the section on an Orientation to Two Competing Imaginaries needed greater clarity and contrast between the two. I reworked that section and the one before it on Emerging Environmental Technologies so that it makes the similarities and differences between these to high-tech interventions clearer.
I did not add anything more to the section where I quote Minteer (about the genetic gap the reviewer suggests), although I found those comments very helpful and a larger project I am thinking about. Mostly, I left that off for reasons of length of the paper.
In general, nearly every section of the paper has now been reworked, and some extensively (or simply cut). Lots of sections have been carefully proofread as well, and clarified.
thanks for this very helpful feedback. Marked up copy attached.